

# Immune fitness and lifestyle habits of Saudi medical students: a cross sectional study

Azzah S. Alharbi

Medical Microbiology and Parasitology Department, Faculty of Medicine, King Abdul Aziz University, Jeddah, Saudi Arabia
Special Infectious Agents Unit, King Fahd Medical Research Center, King Abdulaziz University, Jeddah, Saudi Arabia

## ABSTRACT

**Introduction**. Immune function reaches an optimum level in young adults. However, young adults are more likely to adopt potentially harmful habits that may pose a risk to their long-term health and immune fitness, and which eventually may put a substantial burden on the healthcare system. This study aimed to assess the status of medical students' immune fitness, using the immune status questionnaire (ISQ) and exploring the association with the commonly adopted lifestyle habits hypothesized to have an impact on immune functions.

**Methods**. A descriptive, cross-sectional study was conducted among preclinical students attending the medical school of King Abdulaziz University. An online self-reported questionnaire was used to assess the immune status (ISQ), perceived (momentary) immune fitness, general health, lifestyle habits and students' perception of these lifestyle-associated impacts on immune fitness. Descriptive, Spearman's correlation and stepwise linear regression analyses were performed.

**Results**. In a pooled sample of 211 participants, the overall ISQ score was $6.00 \pm 5.0$ with statistically significant abnormally lower scores in females (ISQ $5.00 \pm 5.0$, $p < 0.001$). 49.29% of respondents experienced poor immune fitness as measured by the ISQ ($<6$). The ISQ score was significantly correlated with fast and fatty food consumption ($p = 0.003$), daytime sleepiness ($p = 0.001$), and BMI subgroups ($p = 0.028$) negatively and positively correlated with adherence to a program of exercise ($p = 0.005$). A total of 41.23% of participants who reported a normal immune health, rated at $\geq 6$ were graded below 6 on the ISQ score. Only 62.6% of students were able to correctly identify the effects of fast and fatty food consumption on immune fitness.

**Conclusion**. Poor immune fitness was common among medical students in KAU and associated significantly with their adopted lifestyle habits. Although, other factors can be significant contributors, biased immune health perception and lack of awareness of these lifestyle-associated impacts on immune fitness and general health may hinder the adoption of healthier habits. Immune biomarkers should be implemented in future work.

Corresponding author
Azzah S. Alharbi,
asalharbi3@kau.edu.sa

## INTRODUCTION

Immune fitness refers to an individual's ability to elicit adequate immune responses efficiently against health challenges, such as infectious pathogens and other internal or external threats, ensuing normal health and better quality of life (*Laupèze et al., 2021*; *van de Loo et al., 2020*). Several lines of evidence suggest that reduced immune fitness plays a fundamental role in the pathogenesis of infections, cancer, depression, neurodegenerative disorders, metabolic and vascular diseases (*Garbarino et al., 2021*). Assessing how well the immune system functions, *i.e.*, immune fitness subjectively, is pivotal to screening for individuals at a higher risk of immune-related disease. Thus, an early cost-effective intervention could eventually be enabled. In addition to genetic aspects and age-related changes, lifestyle-related factors specific to each life stage exert substantial impacts in shaping one's immune fitness and appear to pose a greater impact (*Laupèze et al., 2021*; *Mangino et al., 2017*; *Te Velde et al., 2016*; *Brodin et al., 2015*; *Venter et al., 2020*). They affect immune function directly and/or indirectly by altering the homeostasis of the gut microbiota, a well-known regulator of the immune system (*Te Velde et al., 2016*). What we consume and where or how we live makes us who we are (*Brüssow & Parkinson, 2014*). Consumption of a high-calorie diet, known as the Western diet (WD), rich in saturated fatty acids, refined carbohydrate and deficient in fibres, minerals, vitamins, antioxidants and many other unmodified nutritious elements impairs the host's immune functioning (*Venter et al., 2020*; *Christ, Lauterbach & Latz, 2019*; *Lindseth, 2018*; *González Olmo, Butler & Barrientos, 2021*), potentially via induction of inflammation, increasing risk of infection (*Heras et al., 2022*; *Christ et al., 2018*) and lowering the host's ability to control infection (*Myles, 2014*; *Hyoju et al., 2019*). Saturated fatty acids, (SFAs) a main ingredient of the WD, induces a proinflammatory state by activating macrophage's toll like receptors (TLRs), TLR4 in particular, residing in the gastrointestinal tract (GIT) following absorption. Subsequently, several intracellular singling cascades are activated and lead to the release of many pro inflammatory mediators such as tumour necrosis factor alpha (TNF $\alpha$), interleukin-6 (IL-6), and interferon gamma (INF $\gamma$) (*Christ, Lauterbach & Latz, 2019*; *González Olmo, Butler & Barrientos, 2021*). SFAs induce gut dysbiosis that leads to an accumulation of lipopolysaccharides (LPS), a TLR4 activating ligand (*Christ, Lauterbach & Latz, 2019*; *González Olmo, Butler & Barrientos, 2021*). Even though the most recent *in vitro* studies demonstrate the capability of polyunsaturated fatty acids (PUFAs), another component of the WD, to reduce the activity of immune cells (for example, linoleic acid supressed T cell activation and differentiation and promoted its death (*Hidalgo, Carretta & Burgos, 2021*). Many other ingredients of the WD are expected to negatively influence immune function and are currently being studied (*Christ, Lauterbach & Latz, 2019*). In contrast, a negative correlation between inflammatory markers and a diet rich in natural unprocessed components such as the Mediterranean diet (MD) has been reported (*Christ, Lauterbach & Latz, 2019*). An additional dietary component with reportedly profound impacts on immune functioning is fibre that is found in vegetables, fruits and whole grains. Gut microbiota ferment some of the dietary fibre into short chain FAs (SCFAs) such as propionate, and butyrate. SCFAs exert immunomodulatory effects by supressing

the release of pro inflammatory chemokines and cytokine by dendritic cells and antigen presenting cells (APCs) to maintain immunocompetence (*Nastasi et al., 2015*). Indeed, butyrate has been reported to enhance resistance to enteric pathogens, elicit antibacterial activity in intestinal macrophages *in vivo* (*Schulthess et al., 2019*) stimulate expression of tight junction proteins and thus, diminish intestinal permeability (*Yan & Ajuwon, 2017*). A wide range of micronutrients, such as vitamins A, B6, B12, B9, C, D, E, and trace minerals, like zinc, copper, iron, selenium and magnesium, has been shown to maintain a well-functioning immune system (*Calder et al., 2020*).

Other lifestyle changes associated with a Western dietary pattern including physical inactivity, sleep disturbance, higher exposure to environmental pollution, and heightened amounts of stress may also alter immune function and hence raise the risk of infections and inflammatory disorder (*Laupèze et al., 2021*; *Christ, Lauterbach & Latz, 2019*; *de Frel et al., 2020*). On the other hand, it has been shown that adherence to a program of regular physical workouts maintains efficient immune function (*Furtado et al., 2021*). Physical activity with moderate intensity plays an anti-inflammatory role (*Hojan et al., 2016*), enhancing the function of neutrophils, monocyte, NK, and macrophage and T cell proliferation (*Laupèze et al., 2021*). A higher risk of infection has been observed in individuals with a sedentary life style (*de Frel et al., 2020*). Sleep of adequate duration and sufficient quality has been reported to have an impact on a variety of immunological parameters, being linked to a lower risk of infection, mediating inflammatory homeostasis and improving immunological memory (*de Frel et al., 2020*; *Lange, Born & Westermann, 2019*; *Besedovsky, Lange & Haack, 2019*). Sleep deprivation (*e.g.*, short sleep duration, sleep disturbance) can cause chronic, low-grade inflammation throughout the body, which is linked to inflammatory disorders such as diabetes, atherosclerosis, and neurodegeneration (*Besedovsky, Lange & Haack, 2019*). In sleep deficiency, the reduction in T cell counts (CD4,CD3 and CD8 T cells) and the activity of NK cells, and the increase in various inflammatory cytokines were recorded (*Asif, Iqbal & Nazir, 2017*), so that sleeping less than 6 h a day increases susceptibility to infection and decreases the antibody titer level (*de Frel et al., 2020*). Exposure to environmental pollutants such as smoking affects many cells of both innate (*e.g.*, DCs, macrophages and NK cells) and adaptive immunity (*e.g.*, T helper cells, regulatory T cells, cytotoxic T cells, B cells and memory cells) to attenuate protective and/or exacerbate pathogenic immune responses (*Qiu et al., 2017*). Indirectly, smoking provokes gut dysbiosis characterized by a higher amount of inflammatory microbial composition (*Yan et al., 2021*).

Immune function reaches its optimum in adolescence when young adults ought to be efficiently protected against infections (*Simon, Hollander & McMichael, 2015*). However, they are more likely to adopt new life-style behaviours that adversely influence their immune competency and raise the susceptibility of infection and inflammation (*Christ, Lauterbach & Latz, 2019*; *Maggini, Pierre & Calder, 2018*). At the time of their admission to university, and medical school in particular, students are commonly predisposed to adopting a western dietary pattern due to a lack of time, the easy availability of WD products and the higher prices of nutritious foods (*Sogari et al., 2018*; *Bárbara & Ferreira-Pêgo, 2020*). This dietary pattern is correlated significantly with obesity (*Christ, Lauterbach & Latz, 2019*; *Lindseth,*

*2018*). Additionally, during this period medical students experience anxiety, stress, and poor sleep (*Hershner & Chervin, 2014*; *Schlarb, Friedrich & Clazen, 2017*; *Ibrahim et al., 2017*) as a result of extensive study and the need to manage their time between lectures, exams, work, and social life (*Dragun et al., 2021*). The majority of these health-related lifestyle behaviours are largely at the individual 's control and are quite simple to alter (*Laupèze et al., 2021*). Furthermore, medical students are regarded as role models whose daily life habits and knowledge might affect others' decisions to adapt their lifestyle (*Sondakh et al., 2021*). Thus, it is a legitimate question as to whether students are aware of this immunological alteration driven by lifestyle related factors and to what extent these alterations have an impact on their immune fitness. Therefore, the aim of this study is to assess how strong the immune status (immune fitness) of medical students is as well as to identify the common adult lifestyle factors associated with poor immune status in order to design early preventive strategies.

## MATERIAL AND METHODS

### Study design and participant

A descriptive, cross-sectional study design was employed. A structured, designed questionnaire, prepared by using Google Forms, was distributed online among 2nd and 3rd year pre-clinical students, aged 18 or older attending medical school at King Abdul-Aziz University (KAU), Saudi Arabia, Jeddah. A list of enrolled students was obtained from Academic Affairs. Based on the assumption of a 95% confidence interval and precision of 5%, the required sample size was estimated to be 240. The study protocol was approved by the Research Ethics Committee at KAU, with a Reference Number of 685-20 and conducted in accordance with the Declaration of Helsinki. The survey was anonymous, and participation was voluntary. The aim of the study, and the purpose of data collection were clearly described in the introduction to the questionnaire. Participants' consent was also recorded online before any responses were provided. Subjects who agreed to participate were able to access the questionnaire. Multiple responses per subjects were blocked. Participants with a history of any medical illness or taking medications were excluded from the study. Data were collected between August and November 2021.

### Questionnaire content

Age, gender, height, and weight were among the demographic data collected. Body mass index (BMI) was computed using the following formula: body weight in kg. divided by height in $m^2$. Single-item questions were used to subjectively evaluate current perceived immune and general health status on a scale ranging from 0 (worst) to 10 (best) (*Donners et al., 2015*; *Lantman et al, 2017*; *Abdulahad et al., 2019*; *Kiani et al., 2021*) with a cut-off point at $\geq 6$ regarded as normal status. The student's present perception of diminished immune functioning was evaluated by means of a yes/no question (*Donners et al., 2015*; *Lantman et al, 2017*; *Abdulahad et al., 2019*). The Immune Status Questionnaire (ISQ) was used to evaluate the participant's immune status during the last 12 months (*Abdulahad et al., 2019*; *Kiani et al., 2021*; *Wilod Versprille et al., 2019*). The ISQ is a seven-item scale that rates the frequency of sudden high fever, diarrhoea, headache, skin issues, muscle
and joint discomfort, common cold, and cough on a 5-point Likert scale ranging from never, occasionally, regularly, often, to always. The overall ISQ score ranges from 0 to 10, with a cut-off value of <6 suggesting a lower immunological fitness during the last 12 months. The reliability and validity of the ISQ have been shown in various studies (*Wilod Versprille et al., 2019*). Previous studies have found that the 1-item scores of perceived immune functioning and health are highly correlated with ISQ scores (*Donners et al., 2015*; *Lantman et al, 2017*; *Abdulahad et al., 2019*; *Wilod Versprille et al., 2019*). This ISQ was incorporated into a questionnaire written in clear English which also included a series of straightforward questions about participants' adherence to and awareness of potential immune health-related lifestyle practices. All participants were asked to report the frequency of each of the following lifestyle habits over the past 12 months: consumption of fast food rich in fats and sugar, healthy balanced diet rich in fruits and vegetable and physical activity, the duration of night sleeping, daytime sleepiness and smoking status. Additionally, all study participants were questioned regarding any recent lifestyle changes from their regular habits over the last 12 months. For medical students' awareness, all participant were asked about the association of each lifestyle habits, as well as direction of association, with immune fitness. The English language was used because at KAU all medical students have attained a sufficient level of English proficiency. The questionnaire took approximately five minutes to complete.

## Statistical analysis

The survey data were stored in a MS Excel application, and then analysed using SPSS software, IBM SPSS statistics, version 20 (Chicgao, IL, USA). Continuous variables were presented using descriptive statistics (*e.g.*, mean and standard deviation, median and interquartile range), while categorical variables were reported in frequencies and percentages. The Shapiro–Wilk test, which would reveal a non-normal distribution, was conducted to check normality for each variable. Mann–Whitney $U$ and chi square tests were used to compare the responses of male and female participants. The Mann–Whitney $U$-test (two groups) and the Kruskal–Wallis $H$-test (three groups) were used to compare the mean of the total ISQ score among students with different lifestyle habits. Correlations between the total ISQ scores and the variables were calculated using the Spearman's $\rho$ correlation coefficient. For explanatory analysis, a forward stepwise linear regression was performed on the whole sample to detect and describe the significant influence of the assessed lifestyle variables (independent variables) on the overall immune status (ISQ score, dependent variable). At each step, the variable were chosen based on its highest correlation with the outcome, ISQ score, (*i.e.*, vibrable that has a $p$-value <0.05) and a $p$ value threshold of 0.05 was used to set a limit on the total number of variables included in the final model. All statistical analysis was performed for the whole sample, and again for the men and women separately. All statistical tests were two-tailed. Statistical significance was indicated by $p$-value <0.05.

## RESULTS

The total number of students participated in the survey was 240. Of these, 29 students were excluded. 21 students did not complete the questionnaire while eight students were on medication, which barred their participation. The data of $N = 211$ medical students, (158 (74.9%) females, and 53 (25.1%) males), were included in the analysis. The demographic and other characteristics are displayed in Table 1. Male students are taller and had higher weight and BMIs. Self-rated general health and immune status perceptions' scores were within the normal range, with statistically significant higher scores among male students. A minority of those surveyed ($N = 26$; 12.3%) reported a reduced perceived immunity, of which ($N = 21$; 80.76%) were female. Surprisingly, the ISQ score for the overall sample was at the borderline (median 6.00, IQR 5.0) with a statistically significant abnormally lower score among females (median 5.00, IQR 5.0, $p < 0.001$). $N = 104$ (49.29%) of respondents experienced poor immune fitness, as measured by the ISQ (<6). The most striking result is that $N = 70$ (33.17%) participants who reported normal general health, rated at $\geq 6$ and $N = 87$ (41.23%) who reported normal immune health, rated at $\geq 6$ were graded below 6 at ISQ score Table 2. Significant differences in students ' ISQ scores were observed in fast food consumption ($P = 0.01$), exercise ($P = 0.029$) and daytime sleepiness ($P = 0.001$) habits, with the most prominent differences among female participants Table 3. Expectedly, students who always consume fast food had a lower ISQ score compared to those who sometimes ($p = 0.006$), rarely ($p = 0.04$) and never ($p = 0.041$) consume fast food. Subjects who always practiced exercise had a higher ISQ score compared to those who rarely $p = 0.022$ and never $p = 0.01$ did. Students who reported to have experienced daytime sleepiness were scored lower at ISQ ( $P = 0.001$). Overweight students with BMI>25 had a significantly lower ISQ than normal weight students (BMI 18.5–24.9, $p = 0.026$). By examining the correlation between the ISQ score and potential lifestyle habits affecting immune fitness, a significant negative correlation was found with fast food consumption ( $p = 0.003$), daytime sleepiness ($p = 0.001$) and BMI subgroups ( $p = 0.028$) Table 4. A significant positive correlation was observed with higher adherence to exercise programs ($p = 0.005$). Stepwise linear regression analysis was conducted for all the samples to investigate the possible contribution of the surveyed independent lifestyle habits to the overall ISQ score as an outcome. Fast food consumption, practicing exercise and daytime sleepiness were the most important factors influencing overall immune fitness and will be explored in further studies Table 5. Surprisingly, $N = 132$ (62.6%) students were able to correctly identify the effects of fast food consumption on immune fitness, but a significant lower level of knowledge was observed among male students, $p = 0.024$. However, the majority of students were familiar with the impacts of other tested lifestyle habits on immune fitness status (Fig. 1).

## DISCUSSION

The period of undergraduate study at university acts as a transitional time into adulthood, during which students begin to make their own nutritional and lifestyle decisions and acquire potentially harmful habits (Al-Awwad et al., 2021). Such habits may continue

**Table 1** Characteristics of the study population.

| | Total | Female | Male | p-Value |
|---|---|---|---|---|
| **Demographics** | | | | |
| N (%) | 211 (100) | 158 (74.9) | 53 (25.1) | |
| Age (years) | 20.00 (±2.00) | 20.00 (±1.00) | 20.00 (±2.00) | .001* |
| Height (m) | 1.63 (±0. 13) | 1.59 (±0.08) | 1.74 (±0. 10) | .001* |
| Weight (kg) | 58.00 (±22.00) | 55.00 (±18.00) | 72.00 (±24.00) | .001* |
| **BMI (kg/m$^2$)** | | | | .001* |
| Underweight<18.5 | 21.83 (±7.01) | 21.23 (±6.80) | 24.22 (±8.03) | |
| Normal weight 18.5–24.9 | 38 (18) | 36 (22.8) | 2 (3.8) | |
| Overweight >25 | 108 (51.2) | 79 (50) | 29 (54.7) | |
| | 65 (30.8) | 43 (27.2) | 22 (41.5) | |
| **Perceived health status score** | 7.00 (±3.00) | 7.00 (±3.00) | 8.00 (±3.00) | .001* |
| **Perceived immune functioning score** | 8.00 (±2.00) | 8.00 (±2.00) | 9 (±2.00) | .002* |
| **Perceived reduced immunity%** | | | | .461 |
| Yes | 26 (12.3) | 21 (13.3) | 5 (9.4) | |
| No | 185 (87.7) | 137 (86.7) | 48 (90.6) | |
| **ISQ score** | 6.00 (±5.00) | 5 (±5.00) | 8 (±4.00) | .001* |

Notes.
Data reported as median (IQR) or number (%); Test used = Mann–Whitney $U$-test; Significant differences between female and male ($p < 0.05$) are indicated by *.
Abbreviations: BMI, Body Mass Index; ISQ, Immune Status Questionnaire.

**Table 2** Students perception of general health and immune health in relation to Immune Status Questionnaire scores (ISQ).

| | | Immune Status Questionnaire score (ISQ) | | |
|---|---|---|---|---|
| | | Abnormal (ISQ < 6) | Normal (ISQ ≥6) | Total |
| Perceived general health score | Abnormal <6 | 34 (16.11) | 15 (7.1) | 49 (23.22) |
| | Normal ≥ 6 | 70 (33.17) | 92 (43.6) | 162(76.78) |
| | Total | 104 (49.29) | 107(50.71) | 211 (100) |
| Perceived immune health score | Abnormal <6 | 17 (8.05) | 6 (2.84) | 23 (10.9) |
| | Normal ≥ 6 | 87 (41.23) | 101(47.86) | 188 (89.1) |
| | Total | 104 (49.29) | 107(50.71) | 211 (100) |

Notes.
data presented as number (%).

throughout adulthood, posing a risk to their long-term health as well as immune fitness and may result in an increased risk of disease, treating which can put a substantial burden on the health care system. To the best of our knowledge, this is the first study assessing the status of immune health among medical students, who are supposed to lead by example, using ISQ and exploring the association with commonly adopted lifestyle habits hypothesised to have an impact on immune functions. The study's most intriguing findings concern the immune health status of our participants with a borderline ISQ score. Consistently, borderline ISQ score of 6.2 was also reported among young students from *Verster et al. (2021)*. Females currently have significantly lower ISQ scores than males, suggesting a better immunological status among males than female, which agrees with

Peer J

**Table 3 Modifiable lifestyle characteristics of the study population and the mean differences of the Immune status questionnaire score (ISQ) across different lifestyles.**

| | | Immune Status Questionnaire score (ISQ) | | | | | | | | |
| | | Overall | | | Female | | | Male | | |
| | | N (%) | Mean (±SD) | P-value | N (%) | Mean (±SD) | P-value | N (%) | Mean (±SD) | P-value |
|---|---|---|---|---|---|---|---|---|---|---|
| Fast foods consumption, how often did you eat fast foods in the last 12 months? | Always | 74 (35.1) | 5 (±3.06) | 0.01* | 49 (31) | 4.20 (±2.99) | 0.001** | 25 (47.2) | 6.56 (±2.61) | 0.185 |
| | Sometime | 94 (44.5) | 6.34 (±2.5) | | 74 (46.8) | 5.94 (±2.47) | | 20 (37.7) | 7.80 (±2.09) | |
| | Rare | 39 (18.5) | 6.33(±2.43) | | 31 (19.6) | 5.93 (±2.29) | | 8 (15.1) | 7.87 (±2.47) | |
| | Never | 4 (1.9) | 8.25 (±2.36) | | 4 (2.5) | 8.25 (±2.36) | | 0.0 (0.0) | 0.0 (±0.0) | |
| | Total | 211 (100) | 5.90 (±2.77) | | 158 (100) | 5.46 (±2.74) | | 53 (100) | 7.22 (±2.44) | |
| Fruits and Vegetables, how often did you eat healthy balanced diet rich in fruits and vegetable in the last 12 | Always | 56 (26.5) | 6.23 (±2.71) | 0.585 | 45 (28.5) | 5.75 (±2.49) | 0.722 | 11 (20.8) | 8.18 (±2.78) | 0.047* |
| | Sometime | 90 (42.7) | 5.7 (±2.87) | | 68 (43.0) | 5.16 (±2.79) | | 22 (41.5) | 7.40 (±2.46) | |
| | Rare | 54 (25.6) | 6.05 (±2.76) | | 37 (23.4) | 5.67 (±2.99) | | 17 (32.1) | 6.88 (±2.02) | |
| | Never | 11 (5.2) | 5.09 (±2.42) | | 8 (5.1) | 5.37 (±2.77) | | 3 (5.7) | 4.33 (±1.15) | |
| | Total | 211 (100) | 5.90 (±2.77) | | 185 (100) | 5.46 (±2.74) | | 53 (100) | 7.22 (±2.44) | |
| How did often you practice exercise in the last 12 months? | Always | 31 (14.7) | 6.96 (±2.84) | 0.029* | 18 (11.4) | 6.55 (±2.28) | 0.252 | 13 (24.5) | 7.53 (±3.50) | 0.127 |
| | Sometime | 76 (36) | 6.06 (±2.47) | | 56 (35.4) | 5.50 (±2.37) | | 20 (37.7) | 7.65 (±2.05) | |
| | Rare | 86 (40.8) | 5.63 (±2.88) | | 69 (43.7) | 5.33 (±3.02) | | 17 (32.1) | 6.88 (±1.83) | |
| | Never | 18 (8.5) | 4.66 (±2.84) | | 15 (9.5) | 4.60 (±3.04) | | 3 (5.1) | 5.00 (±2.00) | |
| | Total | 211 (100) | 5.90(±2.77) | | 158 (100) | 5.46 (±2.74) | | 53 (100) | 7.22 (±2.44) | |
| Sleep, Duration of night sleeping in the last 12 months | >6 h | 75 (35.5) | 5.80 (±2.89) | 0.487 | 53 (33.5) | 5.22 (±2.89) | 0.0763 | 22 (41.5) | 7.18 (±2.44) | 0.309 |
| | 4–6 h | 108 (51.2) | 6.11 (±2.68) | | 83 (52.5) | 5.63 (±2.66) | | 25 (47.2) | 7.68 (±2.13) | |
| | <4 h | 28 (13.3) | 5.39 (±2.83) | | 22 (13.9 | 5.36 (±2.78) | | 6 (11.3) | 5.50 (±3.27) | |
| | Total | 211 (100) | 5.90 (±2.77) | | 158 (100) | 5.46 (±2.74) | | 53 (100) | 7.22 (±2.44) | |
| Daytime sleepiness in the last 12 months | Yes | 131 (62.1) | 5.41 (±2.84) | 0.001** | 98 (62.0) | 4.90 (±2.76) | 0.002** | 33 (62.3) | 6.90 (±2.56) | 0.209 |
| | No | 80 (37.9) | 6.71 (±2.48) | | 60 (38.0) | 6.36 (±2.49) | | 20 (37.7) | 7.75 (±2.19) | |
| | Total | 211 (100) | 5.90 (±2.77) | | 158 (100) | 5.46 (±2.74) | | 53 (100) | 7.22 (±2.44) | |
| Smoking, are you a smoker during the last 12 months? | Yes | 26 (12.3) | 6.00 (±2.99) | 0.81 | 12 (7.6) | 5.83 (±2.88) | 0.706 | 14 (26.4) | 6.14 (±3.18) | 0.122 |
| | No | 185 (87.7) | 5.89 (±2.75) | | 146 (92.4) | 5.43 (±2.74) | | 39 (73.6) | 7.61 (±2.03) | |
| | Total | 211 (100) | 5.90 (±2.77) | | 158 (100) | 5.46 ±(2.74) | | 53 (100) | 7.22 (±2.44) | |
| BMI (kg/m$^2$) | Underweight <18.5 | 38 (18) | 5.97 (±2.45) | 0.074 | 36 (22.8) | 5.97 (±2.44) | 0.103 | 2 (3.8) | 7.5 (±3.53) | 0.024* |
| | Normal 18.5-24.9 | 108 (51.2) | 6.23 (±2.83) | | 79 (50) | 5.60 (±2.72) | | 29 (54.7) | 7.93 (±2.43) | |
| | Overweight >25 | 65 (30.8) | 5.27 (±2.78) | | 43 (27.2) | 4.76 (±2.94) | | 22 (41.5) | 6.27 (±2.16) | |
| | Total | 211 (100) | 5.90 (±2.77) | | 158 (100) | 5.46 (±2.74) | | 53(100) | 5.46 (±2.44) | |

**Notes.**

Data reported as mean (SD) or number (%); Test used = Mann–Whitney $U$-test (two groups) and the Kruskal–Wallis H-test (three groups); Significant differences ($p < 0.05$) are indicated by *$P < 0.05$, **$P < 0.00$*

Abbreviations: BMI, Body Mass Index; ISQ, Immune Status Questionnaire; SD, Standard deviation.

**Table 4  Correlations of modifiable lifestyle variables with Immune status questionnaire scores (ISQ).**

| | Correlation with Immune Status Questionnaire score (ISQ) | | | | | |
| | Overall | | Female | | Male | |
| | r | *P*-value | r | *P*-value | r | *P*-value |
| --- | --- | --- | --- | --- | --- | --- |
| Fast food consumption | −0.201[**] | 0.003 | 0.272[**] | 0.001 | 0.250 | 0.71 |
| Healthy balanced diet rich in fruits and vegetables consumption | 0.040 | 0.567 | −0.004 | 0.964 | −0.346[*] | 0.011 |
| Practicing exercise | 0.191[**] | 0.005 | −0.130 | 0.104 | −0.303[*] | 0.027 |
| Duration of night sleeping | 0.018 | 0.795 | 0.025 | 0.758 | −0.041 | 0.771 |
| Daytime sleepiness | −0.219[**] | 0.001 | .252[**] | 0.001 | 0.174 | 0.212 |
| Smoking | −0.017 | 0.811 | −0.030 | 0.707 | 0.214 | 0.123 |
| BMI (kg/m$^2$) | −0.095 | 0.169 | -.182[*] | 0.022 | −0.258 | 0.062 |
| BMI categories | −0.152[*] | 0.028 | 0.109 | 0.171 | 0.375[**] | 0.006 |
| Weight (Kg) | −0.015 | 0.823 | −0.157[*] | 0.049 | −0.196 | 0.159 |

**Notes.**

For each variable, the correlation with the ISQ score was calculated for whole sample, female and male; Test used=Spearman correlation coefficient (r).

[*]$P < 0.05$.

[**]$P < 0.001$.

Abbreviations: BMI, Body Mass Index; ISQ, Immune Status Questionnaire.

**Table 5  Forward stepwise linear regression analysis of the association between lifestyle variables that might affect the immune health and Immune status questionnaire scores (ISQ).**

| Variables | | B | *SE B* | β | 95.0% CI | | R$^2$ | Adjusted R$^2$ |
| | | | | | LL | UL | | |
| --- | --- | --- | --- | --- | --- | --- | --- | --- |
| | | | | | | | .052 | .047[**] |
| Step 1 | (Constant) | 4.369[***] | .490 | | 3.404 | 5.335 | | |
| | Fast foods consumption | −.813[**] | .242 | .227[**] | .336 | 1.290 | | |
| | | | | | | | .086 | .078[**] |
| Step 2 | (Constant) | 3.089[***] | .662 | | 1.783 | 4.395 | | |
| | Fast foods consumption | −.699[**] | .241 | .195[**] | .223 | 1.175 | | |
| | Daytime sleepiness | −1.085[**] | .385 | .190[**] | .325 | 1.844 | | |
| | | | | | | | .107 | .094[*] |
| Step 3 | (Constant) | 4.571[***] | .949 | | 2.700 | 6.441 | | |
| | Fast foods consumption | −.537[*] | .251 | .150[*] | .042 | 1.031 | | |
| | Daytime sleepiness | −1.096[**] | .382 | .192[**] | .343 | 1.849 | | |
| | Exercise practice | .490[*] | .227 | −.149[*] | −.937 | −.043 | | |

**Notes.**

CI, confidence interval; LL, lower limit; UL, upper limit.

[*]$P < 0.05$.

[**]$P < 0.001$.

[***]$P < 0.0001$.

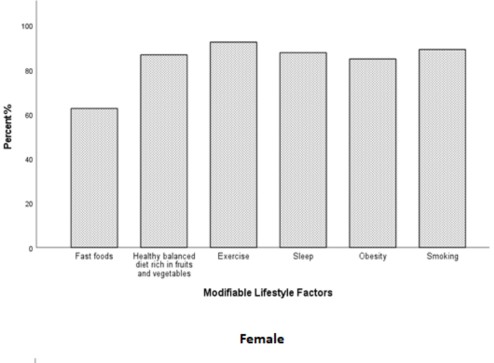

| knowledge level differences' between male and female participants | |
| --- | --- |
| | *P-value* |
| Fast food | 0.024 |
| Healthy balanced diet rich in fruits and vegetables | 0.087 |
| Exercise | 0.783 |
| Sleep | 0.062 |
| Obesity | 0.157 |
| Smoking | 0.009 |

**Note:** test used chi square test. P < 0.05

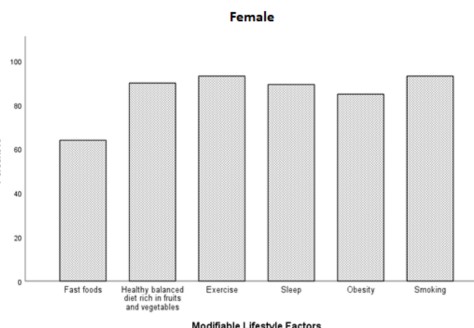

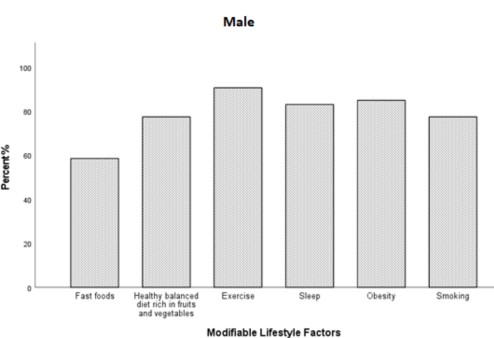

**Figure 1 Medical students' awareness of possible effects of modifiable lifestyle factors on their immune health status.**

previous research from the Netherlands, though a different tool was used (IFQ) (*Lantman et al, 2017*). Females have reported poorer sleep (*Ibrahim et al., 2017*; *Burgard & Ailshire, 2013*), more consumption of fast food (*Yardimci et al., 2012*), and to be more likely to have a sedentary lifestyle (*Varela-Mato et al., 2012*; *Edwards & Sackett, 2016*) anxiety and stress (*Verster et al., 2021*) compared to males, which may explain the observed sex disparities due to the association of these conditions with various aspects of immune system as discussed earlier. This is also apparent in the present study, which found that females reported less physical activity and sleep duration than males. The present results revealed that 49.29% of all medical students had poor immune status and scored below the immune cut-off point (ISQ < 6). This high rate is consistent with a study from the US among the adult population (*Baars et al., 2019*). On the contrary, a lower rate of poor immune status (ISQ<6), 17.5%, was also reported from Saudi Arabia (*Alghamdi et al., 2021a*). This inconsistency might be related to the sample sizes and study population with different ages and sociodemographic backgrounds. The unfavourable immune status detected in this study may be partly related to the adopted lifestyle habits by students. Indeed, several studies support the hypothesis that that there is a link between the immune system and one's lifestyle patterns, which can lead to poor immunological functioning and inflammation as mentioned above. These studies are in line with the present findings, which showed that the ISQ score was correlated negatively with fast food consumption, BMI and daytime sleepiness, with a more pronounced correlation among females. Although a different technique was utilized, this is consistent with prior findings of a strong connection between the human immune

system and consumption of fast food (WD) (*Heras et al., 2022*), BMI (*Pangrazzi et al., 2020*), which was more apparent in women (*Ilavská, Horváthová & Volkovová, 2012*) and sleep deprivation, a contributor of daytime sleepiness (*Tan, Kheirandish-Gozal & Gozal, 2019*). These findings were further confirmed by the stepwise analysis conducted presently that shows the potential contribution of fast food consumption, sleep quality and physical activity respectively. The observed pattern of high fast food intake, physical inactivity, and sleep time less than 6 h reported by a majority of medical students were in concordance with what Alghamdi et al. had reported in their recent study from Saudi Arabia (*Alghamdi et al., 2021b*). Similar findings were observed among medical students from Egypt (*El-Gilany, Abdel-Hady & El Damanawy, 2016*), Jordan (*Yassin et al., 2020*), Nepal (*Sundas et al., 2020*) and other countries (*Mago, Tulsyan & Kour, 2021*; *Usman et al., 2018*; *Almojali et al., 2017*). It might be claimed that medical students should have a better understanding and awareness of healthy lifestyles (*Alghamdi et al., 2021b*; *Vibhute et al., 2018*). However, academic demands, intense study, and challenges throughout the pre-clinical years may make it difficult for students to sustain a healthy lifestyle (*Ibrahim et al., 2017*; *Brehm et al., 2016*). It was worrisome to note that 41.23% of students who rated their perceived immune health at ≥ 6 were actually graded below 6 on the ISQ scale. This is crucial because an individual's self-perception of their general health and immunological status influences their decision to adapt their lifestyle (*Wilod Versprille et al., 2019*). Individuals who believe their health is good are more likely to maintain their existing lifestyle (*Lee et al., 2019*). Participants who overestimate their health always reported a trend of physical inactivity, sleep insufficiency and unhealthy dietary patterns (*Arni et al., 2021*). Such optimistic perception bias of general health may hinder adoption of a healthier lifestyle (*Arni et al., 2021*). Further studies are needed to investigate possible factors contributing to biased health and immunological status perceptions among university students. Students' knowledge level of immune health alterations driven by modifiable lifestyle factors might be another contributor to the unhealthy choices in students. In the present study, only 62.2% of students correctly identified the association of fast-food consumption with immune health status. This may further explain the reported higher percentage of students consuming such products. Consistently, a similar pattern of fast food consumption by Saudi medical students (*Alghamdi et al., 2021b*; *Almutairi et al., 2018*) and college students from Europe (*Chourdakis et al., 2011*) and USA (*McGuire, 2011*) was also reported. On the contrary, 84.8 to 92.4% of the students were well aware of other lifestyle-related influences on immunological health. However, such awareness was not reflected in their healthy lifestyle choices such as physical activity. This finding agrees with what *Alghamdi et al. (2021b)* concluded in their recent study on Saudi students. Indeed, other factors could contribute significantly to students' unhealthy lifestyle choices but understanding of the consequences of such behaviours on immune and general health is required for a lifestyle change to take place. More research is required to investigate whether knowledge and awareness of the unhealthy lifestyle consequences on immune and general health can aid students in adopting healthier choices.

This study provided valuable insight on the status of the immune fitness of medical students, a vulnerable and influential subgroup of the population who contribute

significantly to health promotion. It also supplied an elaborate description of medical students' lifestyle, with several significant features and behaviours in the areas of nutrition, physical activity, and sleep. It investigated the association of these habits with assessed immunological fitness. However, there are certain limitations to the study that should be considered. Firstly, the sample size was relatively small, with the majority of the participants being female. This may be because female students were more concerned about health and lifestyle studies and because of the fact that participation was entirely optional. Secondly, this is a single-institutional study aimed at medical students, the data' generalizability to the population at large is questionable. For different age group and socioeconomic status, the results may differ. Thirdly, this study was cross-sectional in design, thus, association between immunological fitness and lifestyle choices over time is missing. Fourthly, our study's data were purely based on self-reporting. As a result, it is possible that response bias influenced the outcome. Evaluating the association of clinical measures of immune competence (objectively), ISQ score (subjectively) and lifestyle habits should be targeted in future studies to verify the observed relationships.

Considering this study's findings, the lifestyle habits of future medical health professionals should be investigated in depth, and early interventions should be taken. This investigation would reveal more about the factors that affect their professional life as health practitioners and promote the adoption of healthier lifestyles by these prospective physicians.

## CONCLUSION

Poor immune fitness among medical students was prevalent in this study and correlated significantly with fast-food intake, physical inactivity, daytime sleepiness and BMI>25. Biased immune and general health perception and inadequate perception of the potential contributory effects of fast-food intake on the immune health detected in the current study might hinder the adoption of healthier lifestyles. However, adequate perception of the contributary effects of other lifestyle habits was also detected, but not well reflected in their lifestyle choices. Further studies are needed to investigate possible factors contributing to biased health and immunological status perceptions among university students and to find out whether a fundamental understanding of these lifestyle related effects on immune and general health is required for a lifestyle change to take place. The study's findings suggest that future medical health professionals' lifestyle habits should be investigated thoroughly, and early interventions implemented. It is recommended that medical schools should provide a multi-disciplinary team with expertise in health promotion to support a healthy lifestyle among the students and encourage the availability of healthy food and physical activity programs on the campus which may have a positive effect on students' behaviours. Including immune biomarkers would be more valuable in future research.

### Funding

This research was funded by the Deanship of Scientific Research (DSR) at King Abdulaziz University, Jeddah, Saudi Arabia, through institutional funding program, project number (IFPIP: 1013-140-1443). The funders had no role in study design, data collection and analysis, decision to publish, or preparation of the manuscript.

### Grant Disclosures

The following grant information was disclosed by the author:
Deanship of Scientific Research (DSR) at King Abdulaziz University, Jeddah, Saudi Arabia: IFPIP: 1013-140-1443.

### Competing Interests

The author declares that they have no competing interests.

### Author Contributions

- Azzah S. Alharbi conceived and designed the experiments, performed the experiments, analyzed the data, prepared figures and/or tables, authored or reviewed drafts of the article, and approved the final draft.

### Human Ethics

The following information was supplied relating to ethical approvals (i.e., approving body and any reference numbers):

The study protocol was approved by the Research Ethics Committee at KAU, with a Reference Number of 685-20 and conducted in accordance with the Declaration of Helsinki.

### Data Availability

The raw data are available as a Supplementary File.

### Supplemental Information

Supplemental information for this article can be found online at http://dx.doi.org/10.7717/peerj.14363#supplemental-information.

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
