# Peer review of "Immune fitness and lifestyle habits of Saudi medical students: a cross sectional study"

_PeerJ, doi:10.7717/peerj.14363_

## Round 0.1 · original submission · Major Revisions

· Academic Editor

Major Revisions

Dear Authors. Please, note all suggestions and revise or respond accordingly. Regards.

Reviewer 1 ·

Basic reporting

1. In the abstract “… abnormally lower scores in females (ISQ 5.00±5.0, p<0.000).” I suppose it should be “p<0.001”?
2. In the Statistical Analysis section “The Shapiro-Wilk test, which would reveal a non-normal distribution, was conducted to check normality for each variable.” Did you apply any adjustment, e.g., transformation, if the normality assumption didn’t hold?

Experimental design

1. In the Material and Methods section, it first stated that “… the required sample size was estimated to be 240.” but later “Of 700 medical students invited, 240 agreed to participate, …” how exactly was the sample size determined? Please clarify.
2. Also in the Material and Methods section “The Immune Status Questionnaire (ISQ) was used to evaluate the participant’s immune status during the last year” Did the questionnaire measure ISQ score of last year but other aspects of current status? If so, how to assure a fair comparison? Please clarify.

Validity of the findings

No comment.

·

Basic reporting

First of all, I think the introduction is to long, it maybe shorter? I suggest the author goes right to the problem about wost behavior as a western food. Talk slightly on immune metabolism , leave hard explanations to discuss section. I suggest 1 1/2 page to introduction.

Experimental design

Questionnaries made by Google form has a big bias because the Respondent can use the same formulary many times. I suggest the author mention on text that configuration task to block was actived avoid the same voluntary apply the formulary once again.
The author used only one experimental group it is a weak point of study. A control group is necessary to turn it a experimental study
The author talks about fatty food consumption but didn't informed what kind of instrument used to obtain this information, Please show it on method section

Validity of the findings

The finding's validity was committed because it is impossible to affirm the finding without a control group

Additional comments

I encourage author to review the suggests, The medical team have a mission to care but they don't care themselves. So the findings must be showed, and political proccedures must be applyed by university to care the student's health

Reviewer 3 ·

Basic reporting

This is interesting and well-written article. Research background and motivation were clearly stated. The data collection approach and statistical analysis were used appropriately and the study results were described clearly in the manuscript. Two specific questions below.

Line 207-210. The study found striking results that a proportion of participants who reported normal general heath and normal immune health were graded blow 6 at ISQ score. Maybe I missed it, but I did not see the authors discussed this striking results sufficiently in the article. Please elaborate.

For the linear stepwise regression model, it’s not clear how the authors built the model. Based on table 5, it seems the authors used forward selection? Please provide more details how the regression model was built and validated and how the variable selection was performed.

Experimental design

See comments above

Validity of the findings

See comments above

---

## Round 0.2 · accepted · Accept

· Academic Editor

Accept

Congratulations, your manuscript was accepted. Please, follow all steps until the publication. PeerJ appreciates that you have considered our scientific publication and that you have trusted our review process.

Reviewer 1 ·

Basic reporting

No comment.

Experimental design

No comment.

Validity of the findings

No comment.